The effects of afforestation on soil bacterial communities in temperate grassland are modulated by soil chemical properties

Wu Shu-Hong wshuhong@bjfu.edu.cn 1
Huang Bing-Hong 2
Gao Jian 3
Wang Siqi 1
Liao Pei-Chun pcliao@ntnu.edu.tw 2
1 School of Nature Conservation, Beijing Forestry University , Beijing , China
2 Department of Life Science, National Taiwan Normal University , Taipei , Taiwan
3 Faculty of Resources and Environment, Baotou Teachers’ College, Inner Mongolia University of Science and Technology , Inner Mongolia , China
Gonzalez-Meler Miquel
Electronic publication date: 2019 Jan 11
Publication date: 2019
Volume: 7
Electronic Location ID: e6147
Received 2018 May 30; Accepted 2018 Nov 21
Copyright: ©2019 Wu et al.
Copyright year: 2019
Copyright holder: Wu et al.
License: This is an open access article distributed under the terms of the Creative Commons Attribution License, which permits unrestricted use, distribution, reproduction and adaptation in any medium and for any purpose provided that it is properly attributed. For attribution, the original author(s), title, publication source (PeerJ) and either DOI or URL of the article must be cited.
License URL: https://creativecommons.org/licenses/by/4.0/

Keywords: Grassland afforestation, Microbial composition, Microbial ecological function, Soil chemical properties

Funding: Fundamental Research Funds for the Central Universities Ministry of Science and Technology, Taiwan MOST 105-2628-B-003-001-MY3 MOST 105-2628-B-003-002-MY3 This research was financially supported by the Fundamental Research Funds for the Central Universities (to Shu-Hong Wu) and supported by the Ministry of Science and Technology, Taiwan (MOST 105-2628-B-003-001-MY3 and MOST 105-2628-B-003-002-MY3) (to Pei-Chun Liao). This article was also subsidized by the National Taiwan Normal University (NTNU). The funders had no role in study design, data collection and analysis, decision to publish, or preparation of the manuscript.

==============================
Grassland afforestation dramatically affects the abiotic, biotic, and ecological function properties of the original ecosystems. Interference from afforestation might disrupt the stasis of soil physicochemical properties and the dynamic balance of microbiota. Some studies have suggested low sensitivity of soil properties and bacterial community to afforestation, but the apparent lack of a significant relationship is probably due to the confounding effects of the generalist habitat and rare bacterial communities. In this study, soil chemical and prokaryotic properties in a 30-year-old Mongolia pine (Pinus sylvestris var. mongolica Litv.) afforested region and adjacent grassland in Inner Mongolia were classified and quantified. Our results indicate that the high richness of rare microbes accounts for the alpha-diversity of the soil microbiome. Few OTUs of generalist (core bacteria) and habitat-specialist bacteria are present. However, the high abundance of this small number of OTUs governs the beta-diversity of the grassland and afforested land bacterial communities. Afforestation has changed the soil chemical properties, thus indirectly affecting the soil bacterial composition rather than richness. The contents of soil P, Ca2+, and Fe3+ account for differentially abundant OTUs such as Planctomycetes and subsequent changes in the ecologically functional potential of soil bacterial communities due to grassland afforestation. We conclude that grassland afforestation has changed the chemical properties and composition of the soil and ecological functions of the soil bacterial community and that these effects of afforestation on the microbiome have been modulated by changes in soil chemical properties.

Introduction

After a decline in forest coverage in China from roughly 40–60% in remote antiquity (He et al., 2008) to only 8.6% due to war, urban construction, and reclamation, efforts in the last 30 years by the Chinese government to promote afforestation have increased forest coverage to nearly 20% (State Forestry Administration of China, 2011). However, these plantations are not exclusively located at the original sites of deforestation, and consequently, these deforestation and afforestation events have greatly changed the landscape and ecosystems of China (Ahrends et al., 2017). Mongolia pine (Pinus sylvestris var. mongolica Litv.), an endemic tree in Inner Mongolia including in Honghua’erji, is one of the main forestation species in temperate regions of China. Scots pine (P. sylvestris L.), a relative of Mongolia pine that is widespread from Western Europe to Eastern Siberia, is also an important tree in forestry (Krakau et al., 2013) and is physiologically sensitive to environmental pollution (Chudzińska & Diatta, 2014), geographic weather variation (Oleksyn et al., 2003) and climate change (Hurme et al., 1997; Savolainen et al., 2004). Studies have demonstrated that Scots pine and, by reasonable extension, Mongolia pine have broad, plastic adaptability in response to environmental heterogeneity, supporting the wide use of these trees in afforestation.

Grassland afforestation is an artificial and direct change in vegetation that alters both the above- and underground ecosystems, including abiotic changes (Li et al., 2016; Peng et al., 2014; Khamzina, Lamers & Martius, 2016; Lafleur et al., 2015), biotic changes (Ma et al., 2013; Márquez et al., 2015; Pedley et al., 2014; Gunina et al., 2017; Šnajdr et al., 2013; Xiao et al., 2017), and ecological function changes (Lu et al., 2017; Ren et al., 2016; Cibils et al., 2015). Land use is a significant determinant of runoff and soil redistribution processes (Arnáez et al., 2015), and soil microbial species responses may be sensitive to compositional changes in both species and ecological functions (Xiao et al., 2017). However, these underground biological changes are sometimes ascribed to changes in litter amount and chemistry that stimulate the development of the fungal community rather than the bacterial community (Klein et al., 1995). The response of the prokaryotic microbiome to changes in vegetation type is small, likely because bacterial assemblies are based mainly on functional genes rather than species (Burke et al., 2011), and the changes in soil properties due to afforestation are relatively small and occur slowly (Gunina et al., 2017; Jangid et al., 2011). In addition, abundant bacterial “core species” with highly conserved core functions (Falkowski, Fenchel & Delong, 2008) and rarely occurring species (Ai et al., 2013) may act as confounders in statistical analyses of the effects of afforestation or soil properties on prokaryotic microbiome change. Therefore, habitat generalists (core species) and specialists (divergent species) in bacterial communities must be classified before analysis, particularly when using high-throughput sequencing (HTS) technology (e.g., 16S rRNA metagenomic sequencing), which can generate huge amounts of data, to outline the composition and structure of bacterial communities (Székely & Langenheder, 2014).

Although the environmental bacterial composition may be sensitive to environmental variation, the effects of these bacterial composition changes on ecofunction remain unclear. A comprehensive review demonstrated that the soil microbial composition varies to alter acquisition, metabolism, and degradation processes in response to changes in soil phosphorus (P) due to grassland afforestation (Chen, Condron & Xu, 2008). More importantly, environmental conditions and existing microbial diversity determine the ecological function and nutrient transformation efficiency of soil microbiota (Zechmeister-Boltenstern et al., 2015). Therefore, quantifying the effects of afforestation on changes in soil properties, bacterial composition, and ecofunctions would accelerate the understanding of plant-soil-microbe interactions.

In this study, to quantify the effect of afforestation on soil properties and soil bacterial composition and functions, soil chemical and bacterial compositions were measured in an afforested region and adjacent grassland. Based on previous studies that have demonstrated functional assembly of bacterial communities (Burke et al., 2011) and a small response of bacterial communities to afforestation, with low sensitivity to soil properties (Gunina et al., 2017; Jangid et al., 2011; Klein et al., 1995), we hypothesized the following: (1) The soil prokaryotic microbiome is indirectly affected by afforestation mediated by soil chemical properties. If so, soil chemical properties should differ significantly between afforested and grassland sites and should be correlated with the composition of the soil prokaryotic microbiome. (2) Soil bacterial ecological functions are more sensitive to changes in vegetation type than changes in soil bacterial composition. If so, significant differentiation of the predicted ecologically functional potential should be observed between afforested and grassland sites. To test these hypotheses, the 16S rRNA metagenomes of the afforested and adjacent grassland soil bacterial communities were sequenced, and the relative abundances of bacteria and predicted ecophysiological functions were quantified by multivariate and regression analyses. Since root exudates did not significantly affect the bacterial assemblages in a previous study (Wu et al., 2018), we excluded the effect of root exudates in this study. Based on these analyses, we provide a possible explanation of the link between soil chemical properties and bacterial community change in grassland afforestation.

Materials and Methods

Study sites and sampling

The study site was located 6 km west of the town of Honghua’erji in an artificial forest produced by seedling afforestation in a large area of thin grassland (savanna, N48.257443° E119.996448°). The annual precipitation is approximately 390 mm, and the mean annual temperature is roughly −2.4 °C. The elevation is approximately 740–1,100 m. The forest coverage is as high as 69.8% in the forest regions of the nature reserve, while the Honghua’erji grassland is a bare grassland ecosystem that is mainly composed of weedy species such as Stipa baicalensis, Festuca ovina, and Carex pediformis (Wen et al., 2002). The field experiments were approved by the Honghua’erji Nature Reserve (permit number 200/66150221). The tree ages of the Mongolian pines in this afforestation forest range from 27 to 33 years, with a diameter at breast height (DBH) of 21.21 ± 4.19 cm (12.09∼32.10 cm) and tree height of 12.86 ± 0.86 (11.2∼14.6 m).

At the study site, we collected soil pit samples from 20 quadrants, including 10 inside the forest (i.e., forest soils) and 10 in the adjacent grassland (i.e., grassland soils). Each surface area of quadrant was roughly 100 m2. At each location, 1-kg soil samples were collected at a depth of 50 cm in permeable bags of nylon mesh from 10 quadrants located 10 to 100 m from each other (Fig. S1). This depth was chosen because the major microbiome composition and the major transition of soil physicochemical characteristics are located at a soil depth of approximately 50 cm (Fierer, Schimel & Holden, 2003). All equipment was sterilized by an autoclave. The soil samples were separated into two parts: one part was dried for quantification of soil physicochemical properties, and the other was used to quantify the microbiome. The latter samples were stored in RNAguardian stabilization solution (MBGEN Biosciences, Taipei, Taiwan) on ice immediately after collection until transfer to the laboratory, where the samples were stored at −20 °C before metagenomic DNA extraction.

Quantifying soil chemical properties

Organic carbon (C) was measured by the external-heat potassium dichromate oxidation method, and total nitrogen content (N) was measured by the Kjeldahl distillation method. The inductively coupled plasma (ICP) method was used with soils digested in a mixture of HF–HClO4–HNO3 to quantify the contents of the soil elements K+, P, Ca2+, Fe 3+, Mg2+, and Na+. Soil pH was determined using a Sartorius pH meter PB-10 (Germany) with a soil:water ratio of 1:2.5. All soil properties were determined and quantified by the State Key Laboratory of Vegetation and Environmental Change, Institute of Botany, Chinese Academy Sciences.

16S rRNA metagenome sequencing

Bacterial metagenomic DNA was extracted with an EZNA® Soil DNA Kit (Qiagen, Valencia, CA, USA), and the concentration was adjusted to 50 ng ml−1 via dilution. The metagenomic DNA was quantified using Qubit® 2.0 (Invitrogen, Life Technologies, CA, USA). The primers 341F (5′-CCTACGGGNGGCWGCAG-3′) and 805R (5′-GACTACHVGGGTATCTAATCC-3′) were used to amplify the V3-V4 hypervariable 16S rRNA region (Mizrahi-Man, Davenport & Gilad, 2013), and the PCR products were used to construct a DNA library with the Roche GS FLX Titanium emPCR kit (Roche Applied Science). The DNA libraries were then sequenced by Sangon Biotech Co. (Shanghai, China) on an Illumina MiSeq 2X300. The sequencing procedures followed the manufacturer’s instructions.

Before analysis, the raw HTS data were cleaned by removing sequence fragments shorter than 200 bp or with missing barcodes or polyN or polyA/T in the Ribosomal Database Project (RDP) (Cole et al., 2007). We also discarded reads with PHRED quality scores <Q25 (Ewing & Green, 1998; Ewing et al., 1998). The Mothur package was used to remove non-prokaryotic sequences and de-noise and trim the sequences (Mothur ver. 1.30.1, http://www.mothur.org/wiki/MiSeq_SOP (Schloss et al., 2009). Chimeric sequences were removed using Uchime (Edgar et al., 2011). After quality filtering, we clustered sequences using a criterion of >97% sequence similarity as operational taxonomic units (OTUs) defined as representing the same species by the UPARSE pipeline (http://drive5.com/uparse/) (Edgar, 2013) and closed-reference OTU picking. We estimated the relative abundances (RAs) of the soil microbiome according to 16S rRNA metagenome sequencing. The rarefied OTU table was generated using Qiime (Caporaso et al., 2010) and deposited in Mendeley (doi: 10.17632/gjskh8wswz.1). All steps were conducted by Sangon Biotech Co. (Shanghai, China) according to their pipeline (https://www.sangon.com/services_ngs_metseq.html). Each OTU was annotated and classified according to the RDP classifier and SILVA database. The raw sequence data were deposited in NCBI GenBank under Bioproject PRJNA317430 (accession number: SAMN04607375).

Predictive functional profiling of bacterial communities

PICRUSt v. 1.0.0, a functional prediction tool for estimating shared gene content according to the corresponding bacterial phylogeny, was used to predict the functional potential of each sample (Langille et al., 2013). PICRUSt generates the composition of gene families for each metagenome using an extended ancestral-state reconstruction algorithm. The online version of PICRUSt implemented in Galaxy (http://huttenhower.sph.harvard.edu/galaxy/) was used to assist the algorithms. The quality-filtered sequences were assigned to an OTU table against the Greengenes v. 13.5 OTU database (DeSantis et al., 2006) for PICRUSt prediction implemented in QIIME v. 1.8.0 (Caporaso et al., 2010). Each OTU was normalized by its copy number. The functional contribution of each OTU member was reconstructed and predicted by mapping the 16S sequences to their nearest reference genome. ‘Virtual’ metagenomes with gene content abundance were then generated using the Kyoto Encyclopedia of Genes and Genomes (KEGG) Ortholog and Clusters of Orthologous Groups (COGs) databases.

Statistical analyses

All statistical analyses were conducted in R (R Core Team, 2013). This study included two types of measurements, the soil chemical properties and soil bacterial communities (i.e., relative abundances of bacterial OTUs), and one treatment, grassland afforestation (i.e., vegetation type: grassland vs forest). Due to inequality of variance (certain factors such as Na+ and Ca2+ deviated from equal variance in Levene’s test), we first assessed the differences in soil properties between vegetation types by the nonparametric Mann–Whitney U (MW) test using R. In addition, to determine whether the diversity of prokaryotic microbiome differs after afforestation, we calculated diversity indexes such as the Shannon-Wiener H, reciprocal Simpson’s index 1/D, species richness, and Pielou’s evenness for each of the soil bacterial communities and compared them between afforested and grassland sites by the Mann–Whitney U test. We also adopted the R package DESeq2 (Love, Huber & Anders, 2014) to identify OTUs that were differentially abundant between the afforested and grassland sites. A logistic regression model was further used to determine if each of the soil properties predicted the grassland-afforestation treatment. The likelihood ratio test (LRT) was performed to identify the best-fitting model of the logistic regression. To test the hypotheses of a stochastic process of bacterial assembly (random distribution model) or resource-governed assembly (niche-based mechanism), we used Rank-Abundance Dominance (RAD) analysis to display logarithmic relative species abundances against species rank order (McGill et al., 2007). Bray–Curtis distance-based redundancy analysis (dbRDA) implemented in the R package vegan (Dixon & Palmer, 2003) was used to examine the explanatory proportions of the treatment (vegetation type) for these two measurements (soil properties and microbiomes). Because soil chemical properties can also affect the soil bacterial communities (Stutter & Richards, 2012), we further performed a partial dbRDA to assess the effects of grassland afforestation on the soil bacterial communities conditioned by soil chemical properties and the effects of the soil chemical properties on the soil bacterial communities conditioned by vegetation type using the Bray-Curtis distance. The best model were chosen via forward selection by the ordistep function implemented in the R package vegan (Dixon & Palmer, 2003). Type II ANOVA was used to evaluate the fit of the model of each constraint factor. We further identified divergent bacteria (i.e., forest and grassland specialists) using the supermajority rule (>2/3 difference in abundance) with the assistance of the multinomial species classification method (CLAM) test (Chazdon et al., 2011). The soil elements that significantly explained the soil bacterial communities by partial dbRDA were then used as independent factors to predict the abundance of bacterial specialists by the generalized linear model (GLM). We also performed hierarchical cluster analysis (H-cluster) and discriminant analysis of principal components (DAPC) to illustrate the grouping patterns between afforested and grassland sites using the R package adegenet (Jombart, 2008). Because the ecophysiological functions (COGs and KEGG modules) of the soil microbiomes were also predicted, all tests were repeated by replacing the soil bacterial OTUs with the relative abundances of the COGs and KEGG modules. All R codes and input files have been deposited as Mendeley data (https://data.mendeley.com/datasets/gjskh8wswz/1).

Results

Effects of afforestation on soil chemical properties

To characterize the soil chemical properties, the contents of eight elements (organic C, total N, P, K+, Ca2+, Mg2+, Fe3+, and Na+) and the pH value of each soil sample were determined (Table 1). Among these soil variables, the contents of C, P, Ca2+, Mg2+, and Fe3+ were significantly higher in the forest soils than in the grassland soils (P < 0.01), and the pH of the grassland soils was significantly higher than the pH of the forest soils (P = 0.031) (Fig. 1). To estimate the explanatory proportions of the afforestation effect on the variance of soil properties, we performed dbRDA using vegetation type (i.e., forest and grassland) as the independent factor. Vegetation type significantly explained 50.18% of the variance of soil properties (P = 0.002, Table 2 and Fig. S2), indicating that afforestation has changed the soil properties.

Table 1 Contents of soil elements (g kg−1) and pH of the forest and grassland soils reported in format of average ± standard deviation.

Group	C*	N	P*	K+	Ca2+*	Mg2+*	Fe3+*	Na+	pH*	
Forest	12.358 ± 1.950	0.835 ± 0.116	1.469 ± 0.199	26.303 ± 0.341	1.424 ± 0.222	0.456 ± 0.099	8.355 ± 1.047	11.665 ± 0.531	6.216 ± 0.095	
Grassland	7.339 ± 3.025	0.627 ± 0.292	0.990 ± 0.463	26.269 ± 0.563	1.183 ± 0.031	0.313 ± 0.035	5.607 ± 1.308	11.960 ± 0.243	6.318 ± 0.029	
Notes.

* Significant difference between the forest and grassland soils,

If the soil elements are altered after afforestation, the contents of soil elements will predict the vegetation type (i.e., afforestation effect). To test this hypothesis, we compared the simple logistic regression (SLR) model using a single soil element as the independent factor and the multivariate logistic regression (MLR) model using all soil elements as independent factors (i.e., the full model M1) to the empty (null) model M0. Most of the SLR models and M1 rejected the null model M0 (P < 0.005), except the SLR models with K (LRT: P = 0.084) or Na (LRT: P = 0.108) as the independent factor. We further compared the SLR models with each single soil element (C, N, P, Ca2+, Mg2+, Fe3+, and pH) as the independent factor with the MLR model using all C, N, P, Ca2+, Mg2+, Fe3+, and pH as independent factors (full model M2) by LRT to determine which individual factors might be related to afforestation. The SLR models with Mg or Fe as the independent factor could not be rejected by M2 (P = 0.278 and 0.130, respectively, Fig. 2). Although the other SLR models with C, N, P, Ca2+, or pH as the independent factor were rejected by M2, none of the independent factors in M2 could significantly predict the presence of forestation (Z < 10−6, P > 0.9999 in each term). These results indicated that the contents of Mg2+ (0.456 ± 0.099 and 0.313 ± 0.035 g. kg−1 in forest and grassland soils, respectively) and Fe3+ (8.355 ±1.047 and 5.607 ± 1.308 g. kg−1 in forest and grassland soils, respectively) could singly reflect the changes in soil properties due to afforestation (Table 1 and Fig. 2).

Figure 1 Mann–Whitney U test revealing significant differences in soil properties, i.e., contents of organic C, P, Ca2+, Mg2+ and Fe2+ and pH, between the forest (Fr) and grassland (Gr) soils.

Table 2 Summary of the (partial) dbRDA results for the afforestation effect (i.e., vegetation type) on soil properties and on the relative abundance (RA) of soil bacteria.

The effect of soil properties on the bacterial RA was also tested. Type II ANOVA was used to test the model fitting for each independent variable. Significant P values (<0.05) are in bold.

	dbRDA	ANOVA	
	S.S.	Proportion	F	P	
Soil property ∼ Vegetation type	
Constrained	0.027	0.+	18.13	0.002	
Unconstrained	0.027	0.498			
Microbial RA ∼ Vegetation type	
Constrained	0.849	0.163	3.504	0.001	
Unconstrained	4.362	0.837			
Microbial RA ∼ Vegetation type + Condition (Soil property)	
Conditional	3.356	0.644			
Constrained	0.194	0.037	1.053	0.466	
Unconstrained	1.661	0.319			
Microbial RA ∼ Soil property	
Constrained	3.356	0.644			
C	0.595	0.114	3.210	0.001	
P	0.524	0.100	2.823	0.001	
Ca2+	0.467	0.090	2.516	0.001	
Mg2+	0.368	0.071	1.986	0.003	
Fe3+	0.546	0.105	2.940	0.001	
pH	0.228	0.044	1.232	0.145	
N	0.249	0.048	1.343	0.070	
K+	0.188	0.036	1.012	0.438	
Na+	0.190	0.037	1.026	0.365	
Unconstrained	1.855	0.356			
Microbial RA ∼ Soil property + Condition (Vegetation type)	
Conditional	0.849	0.163			
Constrained	2.701	0.518			
C	0.350	0.067	1.899	0.005	
P	0.493	0.095	2.673	0.001	
Ca2+	0.457	0.088	2.477	0.001	
Mg2+	0.350	0.067	1.896	0.004	
Fe3+	0.283	0.054	1.534	0.009	
pH	0.203	0.039	1.102	0.238	
N	0.183	0.035	0.994	0.490	
K+	0.190	0.037	1.031	0.362	
Na+	0.190	0.036	1.029	0.378	
Unconstrained	1.661	0.319			

Figure 2 A simple logistic regression model revealing significant prediction of the absence (0) and presence (1) of forestation based on the contents of the soil elements (A–H) and pH (I).

(A) Organic C; (B) N; (C) P; (D) K; (E) Ca; (F) Fe; (G) Mg; (H) Na); (I) pH.

Afforestation effects on the relative abundances of soil bacteria

Overall, a total of 694,993 sequences were obtained after rarefaction analysis (314,923 for grassland and 380,070 for forest). Divergence of the soil microbiome was inferred by significant or marginal differences in the diversity indices (Shannon–Wiener index H, W = 26, P = 0.0753; Reciprocal Simpson’s index 1/D: W = 23, P = 0.0432; Pielou’s evenness J: W = 8.5, P = 0.0019, Table 3). However, no difference was estimated in the species richness of the soil microbiome between forest and grassland (W = 50, P = 1, Table 3). Hence, we used RA as an indicator to compare the microbiomes of the soil samples (Fig. S3). H-cluster and DAPC (five first PCs of PCA used, which conserved 70% of the variance of the bacterial RA) presented similar patterns of clear divergence of soil microbiomes between the forest and grassland (Fig. 3). A total of 350 OTUs with an adjusted P-value <0.01 exhibited twofold changes between forest and grassland vegetation (178 OTUs were higher in grassland, while 172 were higher in forest). The differentially abundant OTUs were only from the most abundant phyla (Proteobacteria, Acidobacteria, and Verrucomicrobia) and Planctomycetes. These results confirm that the soil bacterial composition has changed due to grassland afforestation despite no change in bacterial richness.

Because the close-reference OTU picking strategy discards samples that do not bin with 97% similarity, assignments of different OTUs were rarely found among different samples. We performed the CLAM test (Chazdon et al., 2011) to classify these OTUs as four types of bacteria using the supermajority (2/3) rule: too-rare bacteria, generalist bacteria, and grassland- and forest-specialist bacteria. In this classification, a large proportion of OTUs (95.46%) belonged to the too-rare bacteria, which accounted for 38.42% and 36.73% of the abundance of grassland and forest soil bacteria, respectively; only 2.98% of the OTUs were generalists, but they accounted for 35.55% and 35.88% of the abundance of grassland and forest soil bacteria, respectively. Only 1.56% of the OTUs belonged to specialists, of which 1061 (0.7%) and 1224 (0.8%) bacterial OTUs were identified as forest and grassland specialists, respectively. The RA of grassland-specialist bacteria was 22.84% in grassland soils and 2.84% in forest soils, while the RA of forest-specialist bacteria was 3.19% in grassland soils and 24.55% in forest soils.

Table 3 Diversity indices comparisons of soil bacterial communities.

	Forest (mean ± SD)	Grassland (mean ± SD)	KW χ2	P	
Shannon–Wiener H	8.429 ± 0.244	8.612 ± 0.174	3.291	0.070	
Reciprocal Simpson’s index 1/D	877.783 ± 281.326	1,190.206 ± 364.143	4.166	0.041	
Species richness (S)	14,685.0 ± 4,846.82	13,664.6 ± 2,107.68	0	1	
Pielou’s evenness (J)	0.883 ± 0.017	0.905 ± 0.008	9.864	0.002	

Figure 3 Divergence of the microbiome between the forest and grassland soils as revealed by (A) hierarchical cluster analysis and (B) discriminant analysis of principal components (DAPC).

We further used the RAs of the bacterial OTUs as the dependent response to access the impact of afforestation on the soil bacterial communities. When vegetation type was used as the categorical independent factor, 16.3% of the variance of bacterial RA was explained significantly (P = 0.001). However, the significant explanatory effect of vegetation type was lost after removing (conditioning on) soil properties (P = 0.466, Table 2). When soil properties were used as a constraint factor, 64.4% of the variance of bacterial RA was explained by soil properties, in which organic C (11.4% explanation), P (10.0%), Ca2+ (9.0%), Mg2+ (7.1%), and Fe3+ (10.5%) significantly fit the model by type II ANOVA (Table 2). Similar significant explanations by C (6.7%), P (9.5%), Ca2+ (8.8%), Mg2+ (6.7%), and Fe3+ (5.4%) were obtained when the effect of vegetation type was removed (i.e., partial dbRDA, Table 2). In addition, when conducting forward model selection, the bacterial RA was best explained by vegetation type, Ca2+, Mg2+, Na+ and pH (constrained proportion: 48.07%). However, although pH is important in shaping the soil microbiome at continental scales (Lauber et al., 2009), the pH did not vary among vegetation types (forest: 6.3–6.36, grassland: 6.13–6.35) in our fine-scale study. There was no collinearity of pH with other factors [variance inflation factor (VIF): 1.125]. Therefore, we do not discuss the importance of pH in our study. These analyses suggested that afforestation has changed the soil chemical properties, with changes in at least Ca2+ and Mg2+ in all analyses, thus indirectly affecting the soil bacterial communities (Figs. S2 and S3).

Effects of soil properties on the divergence of bacterial phyla between forest and grassland soils

To identify the divergent soil bacteria, the habitat-specialist bacterial phyla were classified under the supermajority rule (i.e., 2/3 majority). At the phylum level, five phyla were habitat specialists, including one forest specialist (Thaumarchaeota, phylum RA: 2.3%) and four grassland specialists (Chloroflexi, Fibrobacteres, Nitrospirae, and Parcubacteria, 9.3%). The forest-specialist phylum Thaumarchaeota belongs to Archaea, while the four grassland specialists are Eubacteria.

We further tested the effects of soil properties on the abundance of these habitat specialists by GLM. Because organic C, P, Ca2+, Mg2+, and Fe3+ significantly explained the RA of soil bacteria in dbRDA, these five soil variables were used as independent predictors in GLM. The variances of the bacterial abundance of the sampled soils were significantly or marginally significantly greater than the means, suggesting overdispersion of these responses (P = 0.056, 0.001, 0.051, 0.002, and 0.002; alpha = 52.675, 25.871, 1.065, 23.101, and 76.612 in Thaumarchaeota, Chloroflexi, Fibrobacteres, Nitrospirae, and Parcubacteria, respectively). Hence, we used the quasi-Poisson model for GLM, which led to the same coefficient estimates as the standard Poisson model but with adjustment of the dispersion parameter for overdispersion. Under the quasi-Poisson model, Fe3+ content was marginally correlated with the abundance of the forest specialist Thaumarchaeota; for the grassland specialists, the P content was correlated with the abundances of all four phyla (Table 4). In addition, C, Ca2+, and Mg2+ were significantly correlated with Chloroflexi, Fe3+ was significantly correlated with Nitrospirae, and Ca2+ and Mg2+ were marginally correlated with Parcubacteria (Table 4).

Table 4 Effect of soil properties on the abundance of divergent soil microbial phyla inferred by the generalized linear model (GLM).

	Thaumarchaeota	Chloroflexi	Fibrobacteres	Nitrospirae	Parcubacteria	
	t value	P	t value	P	t value	P	t value	P	t value	P	
Intercept	0.068	0.947	−3.280	0.005	−1.087	0.295	−1.176	0.259	−1.590	0.134	
C	−0.288	0.778	3.501	0.004	0.632	0.538	1.471	0.164	1.714	0.109	
P	−0.420	0.681	4.848	0.0003	3.735	0.002	4.108	0.001	3.326	0.005	
Ca2+	−0.083	0.935	3.857	0.002	1.267	0.226	1.667	0.118	2.007	0.065	
Mg2+	0.031	0.976	−3.727	0.002	−1.174	0.260	−1.374	0.191	−1.934	0.074	
Fe3+	2.088	0.056	−0.332	0.745	−1.263	0.227	−2.900	0.012	−0.575	0.574	
Notes.

Significant P values (<0.05) are in bold.

Afforestation effects on the soil ecophysiological functions predicted by soil microbiome

To understand the ecophysiological functions of the soil bacterial communities, we predicted their functional composition using 16S rRNA gene and databases of reference genomes. A total of 4,659 clusters of orthologous groups (COGs) and 306 KEGG modules (Level-3 KEGG orthology) were identified. Similar to the analyses for testing the effects of afforestation and soil properties on the soil bacterial communities, we used vegetation type and soil elements as predictors to test the explanatory proportion and significance of each predictor on the RAs of the COGs and KEGG modules. dbRDA indicated that 22.0% and 21.2% of the variation of COGs and KEGG modules was significantly explained by vegetation type, respectively, whereas the explanatory proportion decreased to 3.4% and 4.5% when conditioning the soil-property effect (Table 5). Soil properties explained 83.0% and 82.4% of the variation of the COGs and KEGG modules in dbRDA, respectively, in which C, P, Ca2+, Mg2+, Fe3+, and N significantly or marginally fit the model according to type II ANOVA (Table 5). When conditioned on vegetation type, the explanatory proportion decreased slightly to 64.4% and 65.8% for the COGs and KEGG modules, respectively, and the remaining significant fitting factors were P, Ca2+, and Fe3+ (Table 5).

Table 5 Summary of the (partial) dbRDA results for the afforestation effect (i.e., vegetation type) and soil-property effect on the ecological function (COGs and KEGG modules) estimated from the soil bacterial communities.

Type II ANOVA was used to test the model fitting for each independent variable.

	dbRDA	ANOVA	dbRDA	ANOVA	
	S.S.	Proportion	F	P	S.S.	Proportion	F	P	
	COGs ∼ Vegetation type	KEGG ∼ Vegetation type	
Constrained	0.005	0.220	5.081	0.001	0.001	0.212	4.834	0.001	
Unconstrained	0.016	0.780			0.002	0.788			
	COGs ∼ Vegetation type + Condition (Soil property)	KEGG ∼ Vegetation type + Condition (Soil property)	
Conditional	0.017	0.830			0.003	0.824			
Constrained	0.001	0.034	2.267	0.078	1E–04	0.045	3.140	0.038	
Unconstrained	0.003	0.136			4E–04	0.130			
	COGs ∼ Soil property	KEGG ∼ Soil property	
Constrained	0.017	0.830			0.003	0.824			
C	0.002	0.094	5.541	0.001	3E–04	0.097	5.5051	0.001	
P	0.003	0.140	8.234	0.001	0.001	0.165	9.3958	0.001	
Ca	0.003	0.140	8.247	0.001	3E–04	0.107	6.0938	0.001	
Mg	0.002	0.072	4.252	0.005	1E–04	0.042	2.4047	0.063	
Fe	0.002	0.073	4.273	0.010	2E–04	0.066	3.738	0.013	
pH	5E–04	0.023	1.347	0.283	9E–05	0.028	1.5886	0.177	
N	0.005	0.258	15.157	0.001	0.001	0.292	16.615	0.001	
K	2E–04	0.010	0.575	0.653	4E–05	0.012	0.6817	0.580	
Na	4E–04	0.020	1.174	0.312	5E–05	0.016	0.914	0.453	
Unconstrained	0.004	0.170			0.001	0.176			
	COGs ∼ Soil property + Condition (Vegetation type)	KEGG ∼ Soil property + Condition (Vegetation type)	
Conditional	0.005	0.220			0.001	0.212			
Constrained	0.013	0.644			0.002	0.658			
C	0.001	0.027	1.776	0.152	7E–05	0.021	1.478	0.257	
P	0.004	0.172	11.371	0.001	5E–04	0.157	10.877	0.001	
Ca	0.001	0.071	4.723	0.006	2E–04	0.061	4.226	0.018	
Mg	0.001	0.030	1.963	0.133	8E–05	0.027	1.869	0.175	
Fe	0.005	0.251	16.630	0.001	0.001	0.264	18.238	0.001	
pH	0.001	0.032	2.094	0.110	1E–04	0.038	2.650	0.066	
N	4E–04	0.018	1.162	0.302	9E–05	0.028	1.946	0.130	
K	3E–04	0.014	0.928	0.406	7E–05	0.022	1.510	0.223	
Na	0.001	0.030	2.017	0.141	1E–04	0.039	2.691	0.070	
Unconstrained	0.003	0.136			4E–04	0.130			
Notes.

Significant P values (<0.05) are in bold.

Among all COGs, 44 and 21 were identified as forest- and grassland-predominant COG categories by CLAM analysis, respectively. Most of the forest-predominant categories are involved in post-translational regulation, such as ubiquitin or E3 ligase, while many categories in grassland are membrane proteins or are involved in cell transport. By contrast, among all KEGG modules, only two forest-predominant categories and no grassland-predominant categories were identified. Further testing of the correlation of soil elements with these specialist COGs and KEGG modules revealed that 50 of the 65 COGs and both KEGG modules were significantly correlated with at least one soil element under Poisson or quasi-Poisson regression. These significant correlations indicate that soil properties, particularly P, Ca2+, and Fe3+, account for the changes in ecophysiological functional due to grassland afforestation.

Discussion

The closure of the tree canopy and increased litter accumulation that accompany the ecosystem change from grassland to forest may directly alter the soil environment (Bond & Midgley, 2012; Cunningham et al., 2015). The significantly higher contents of soil chemical factors (C, P, Ca 2+, Mg2+, Fe3+, and pH) in the Mongolian pine plantation areas than in the unplanted region suggest a great influence of grassland afforestation on secondary salinization. Soil mineral elements are usually increased in tree-plantation regions where groundwater is insufficient to meet water requirements (Nosetto et al., 2008). High contents of soil elements in a forest suggest not only a larger amount of litter biomass but also a rapid decomposition rate of pine litters compared to other broadleaf flora (Berger et al., 2015). However, despite significant differences in the contents of soil elements between forest and grassland, these soil elements, except Mg2+ and Fe3+ (Fig. 2), could not singly predict the ecosystem change due to afforestation by logistic regression analysis.

Significantly high predictable contents of Fe3+ and Mg2+ in forest soils reflect the characteristics of litter and humus accumulation in forests (Song et al., 2008). Litter decomposition accelerates the conversion and accumulation of soil non-organic elements (Fenchel, King & Blackburn, 2012b). In 2nd-to-3rd-year needle litters of P. sylvestris L., a decrease in the rate of biomass loss but an increase in the release of Fe3+ and Mg2+ were recorded (De Marco et al., 2007). Consequently, we suggest that the high contents of Fe3+ and Mg2+ in the forest soils of our study sites are due to the long, steady accumulation and decomposition of needle litter. The soil element cycling affects and is affected by the composition of the soil microbiota (Fenchel, King & Blackburn, 2012b). For example, the iron bacteria family Comamonadaceae, which was represented by the genera Delftia, Comamonas, Acidovorax, and Albidiferax in our sampling, are able to deposit iron metal oxides under natural conditions. These bacteria were present in soil for both vegetation types. They were slightly more abundant in forest samples and can grow rapidly in iron-rich and acidic substrates (Emerson et al., 2015; Fenchel, King & Blackburn, 2012a). These results indicate that alteration of these chemical properties may lead to a change in the composition of the soil prokaryotic microbiome.

The distributions of the soil bacterial abundances in our samples best fit to Zipf and Zipf–Mandelbrot rank abundance models (Table 6 and Fig. S4 ). The Zipf and Zipf–Mandelbrot rank abundance models belong to the family of random-branching processes; these models suggest that individuals are always derived from ancestor individuals (McGill et al., 2007) and that microbial community assembly is explained by the niche-based mechanism (McGill et al., 2007; Mendes et al., 2014). These models indicate that decades of grassland afforestation have generated soil properties that provide a divergent but stable resource supply for the soil bacterial community, although a strong effect of depth on the structure of bacterial communities should be recognized (Eilers et al., 2012).

Table 6 Deviance of the species-rank abundance distribution (RAD) models revealing the best fits of the Mandelbrot or Zipf-Mandelbrot model for all sampled soil microbial communities.

	Gr01	Gr02	Gr03	Gr04	Gr05	Gr06	Gr07	Gr08	Gr09	Gr10	
Null	23,473.2	24,412.6	20,882.8	21,609.8	29,653.7	23,972.9	16,175.0	21,291.9	25,119.2	31,461.4	
Preemption	28,921.1	30,018.9	25,494.1	26,779.2	36,675.8	29,667.9	19,209.2	26,001.7	30,809.4	38,768.3	
Lognormal	9,527.5	11,539.7	9,701.5	10,157.8	12,942.7	10,585.8	7,802.7	9,847.3	10,650.6	13,265.6	
Zipf	1,652.7	1,201.5	1,168.1	1,010.2	1,722.0	1,421.0	1,016.5	1,169.2	2,228.9	2,215.5	
Mandelbrot	1,057.1	1,201.5	1,010.8	1,010.2	1,525.2	1,165.6	919.3	960.8	1,050.2	1,338.9	
	Fr01	Fr02	Fr03	Fr04	Fr05	Fr06	Fr07	Fr08	Fr09	Fr10	
Null	16,276.4	37,243.8	90,272.5	29,118.5	45,159.8	20,701.9	30,670.6	30,757.4	24,744.7	56,299.3	
Preemption	19,814.6	46,031.3	109,349.0	36,030.8	55,651.5	25,626.1	37,854.4	38,189.6	30,699.1	68,945.4	
Lognormal	7,617.4	15,371.2	30,465.1	12,328.5	18,347.1	9,072.1	12,376.0	12,888.0	10,492.1	21,309.7	
Zipf	1,299.5	2,431.4	4,527.6	1,944.1	3,573.2	1,244.4	1,894.5	1,351.4	1,220.3	2,186.6	
Mandelbrot	867.4	1,699.7	3,043.8	1,248.1	2,080.9	883.3	1,245.9	1,285.5	1,092.0	1,897.3	
Notes.

The lowest Akaike Information Criterion (AIC) values representing the best fit model are shown in bold.

Classification by the supermajority rule revealed that generalist bacteria represented more than 2/3 of the total counts but <3% of bacterial OTUs. The low richness but relatively high abundance of generalists suggests that a great proportion of residents utilize broad resources or are highly tolerant of the environment (Verberk, 2011). It has been suggested that microbes that are present in all or the majority of microbial communities with high abundance represent the core set of genes responsible for key elements of most metabolic pathways (Falkowski, Fenchel & Delong, 2008). Similarly, specialist bacteria exhibited <1% richness but accounted for approximately 1/5 to 1/4 of the RA in the grassland and forest soils. These specialist OTUs with low richness and high abundance are probably more susceptible than generalists to environmental change. Since the original vegetation was scattered grasses, the grassland specialists rarely found in forest soils were those selected against by the afforestation effect; by contrast, forest specialists should be enriched after forestation. The environmental differences (e.g., the contents of soil C, P, Ca2+, Mg2+, and Fe3+, Table 2) could result in resource (niche) divergence to differentiate the bacterial composition descended from the original bacterial communities, reflecting the bacterial abundance distribution in the Zipf and Zipf–Mandelbrot models (Table 6).

In particular, the Fe3+ content was significantly correlated with the abundance of the forest-specialist Archaea phylum Thaumarchaeota, and P was correlated with the four grassland-specialist Eubacteria phyla Chloroflexi, Fibrobacteres, Nitrospirae, and Parcubacteria (Table 4), suggesting that these two soil elements are key factors differentiating soil bacterial composition. Thaumarchaeota encodes the genes ammonia monooxygenase A (amoA, encoding subunit A of AMO) and amoB, which are distantly related to one another and similar to the bacterial amo gene for ammonia oxidation (Stieglmeier, Alves & Schleper, 2014). The high abundance of amoA and its transcripts suggests that ammonia-oxidizing archaea (AOA) are present in higher numbers than ammonia-oxidizing bacteria (Shen et al., 2008) and that nitrogen cycling is enhanced in the forest (Konneke et al., 2005; Stieglmeier, Alves & Schleper, 2014). A high abundance of AOA with a high content of Fe3+ (e.g., ferrate, an ammonia oxidation reagent) could accelerate ammonia oxidation (Sharma, Bloom & Joshi, 1998).

Soil P is closely related to plant growth (Shen et al., 2011), but microbes that transform P to improve uptake by plants are also influenced by both plant species and soil type (Chen, Condron & Xu, 2008). P metabolism is often related to the fungal community (Beever & Burns, 1981; Rodŕıguez & Fraga, 1999). In this study, the specialists that were correlated with soil P content were primarily not responsible for P transformation but were light and aerobic thermophiles (e.g., Chloroflexi), symbionts within ruminant animals (e.g., Fibrobacteres), or involved in the nitrogen cycle (e.g., Nitrospirae). Although not directly linked to P content, these bacterial ecological function reflect the differences between grassland and forest ecosystems. The major changes in the canopy, amounts of litterfall, plant composition, and root-bacteria interactions due to grassland afforestation may explain the differences in soil P content as well as the abundances of these bacterial phyla (Chen, Condron & Xu, 2008; Li, Lee Allen & Wollum, 2004).

By contrast, the significant differences in H, 1/D, and J but not richness between the grassland and forest soil microbiomes suggest that the change in the soil microbiome is due to a difference in species RA rather than species number. A high proportion of “too-rare” bacterial OTUs (95% richness) accounted for >1/3 of the RA of soil bacteria, reflecting transient changes in bacteria in the environments. This rarity could result from stochasticity (Ai et al., 2013), fitness trade-off (Gobet et al., 2012; Gudelj et al., 2010), or biological interactions (García-Fernández, De Marsac & Diez, 2004; Narisawa et al., 2008; Schluter et al., 2015). These rare bacteria are still relevant in ecological functions, including bacterial community assembly and function and biogeochemical cycling (Jousset et al., 2017). The high richness of rare bacteria contributes to the alpha-diversity of the soil bacterial communities.

Based on the dbRDA of the relationships among vegetation type, soil properties, and bacterial composition, we concluded that grassland afforestation has affected the soil chemical properties (Table 2), soil microbiome composition and ecologically functional potential. Post-translational systems may increase the heterogeneity of the soil prokaryotic microbiome and facilitate the coexistence of different bacterial species via functional regulation (Spallek, Robatzek & Göhre, 2009). In addition, these effects of afforestation on the microbiome were modulated by changes in soil chemical properties (Tables 2 and 5). The changes in chemical properties led to differential abundance of low-taxonomic-level OTUs. For example, Planctomycetes, which was differentially abundant between forest and grassland in our data, are sensitive to changes in soil management and physicochemical properties (Buckley et al., 2006). This conclusion was reached because the explanation of soil microbiome by vegetation type decreased or was even lost when conditioned on soil properties (Table 2). Albeit indirectly, afforestation indeed altered the ecological function of the soil microbiome (Table 5).

As discussed above, soil chemical properties interact with the soil bacterial communities. Because the original vegetation before forestation was grassland, the changes in the soil properties are probably attributable to the vegetation changes produced by afforestation. Several studies have suggested that afforestation can influence biotic and abiotic changes in micro- and macro-ecosystems (Jousset et al., 2017; Nosetto et al., 2008; Wang, Zhu & Chen, 2016; Zheng et al., 2017). Here, we suggest that the underground biotic change was indirectly affected by forestation mediated by soil property changes (Table 2), especially the contents of soil P, Ca2+, and Fe3+, which are further related to ecological function changes in soil microbiomes for different vegetation types (Table 5 and Table S1 ). These results are similar to the bacterial abundance changes and compositional shifts reported for a long-term poplar plantation, which were suggested to be highly correlated with the changes in soil properties caused by afforestation (Zheng et al., 2017). Soil bacterial composition has been suggested to be more closely related to plant diversity-controlled abiotic soil properties because of the highly resilient characteristics of bacterial communities due to their fast life cycle (De Vries et al., 2012; Lange et al., 2014).

Conclusions

The change in vegetation type was linked, at least in part, to potential ecological function changes in the soil bacterial communities, despite an indirect impact on bacterial composition and the calculation of the ecologically functional potential from the bacterial composition. Aboveground changes such as abiotic factors and biotic activities may be responsible for these changes in underground ecological functions. According to our study, afforestation may change the soil contents and subsequently affect the ecological functions and alpha- and beta-diversity of bacterial communities. However, the relatively small proportion of bacterial specialists and high proportion of bacterial generalists with respect to ecological function among the bacterial OTUs indicates that the vegetation change preserved a high proportion of the core functions of the soils (Li, Lee Allen & Wollum, 2004). This preservation occurred because the core functional genes were distributed widely across a variety of bacterial taxa. However, the high proportion of functions that were correlated with changes in soil properties indicates that bacterial ecological functions are highly sensitive to environmental change. Future work could focus on the link between anthropogenic actions on vegetation and indicator species such as Planctomycetes or iron bacteria such as Comamonadaceae. These links could be validated by quantitative analyses (e.g., qPCR) to determine how changes in the structure of key bacteria lead to ecological function changes after land cover change.

Supplemental Information

Figure S1 Distribution of forest (blue dot) and grassland (red dot) plot

Map data ©2018 Google.

Click here for additional data file.

Figure S2 dbRAD scatter plot shows the distribution of prokaryotic microbiome composition differences between forest and grassland sites

Click here for additional data file.

Figure S3 Relative abundance of the each bacterial phylum among samples

Click here for additional data file.

Figure S4 Rank-abundance dominance (RAD) model tests showing that the sampled microbial communities were best fit to the Zipf or Zipf–Mandelbrot model

The blue dots and the line in each panel are the observed values and the expectations simulated from the observed data, respectively. Gr and Fr indicate samples from the grassland and forest soils, respectively. The Zipf and Zipf–Mandelbrot models are both niche-based models. Rejection of the null model indicates rejection of the hypothesis of a stochastic process of microbial assembly.

Click here for additional data file.

Table S1 Correlations of the COGs and KEGG modules predicted by the soil microbiome with the soil elements P, Ca, and Fe that significantly explain the variance of the soil microbiome from dbRDA by the generalized linear model (GLM)

A quasi-Poisson distribution model was used for overdispersed COGs.

Click here for additional data file.

We thank Dr. Dawn Schmidt for English editing of the manuscript.

Additional Information and Declarations

Competing Interests

Author Contributions

Field Study Permissions

Data Availability

The authors declare there are no competing interests.

Shu-Hong Wu conceived and designed the experiments, contributed reagents/materials/analysis tools, approved the final draft.

Bing-Hong Huang performed the experiments, analyzed the data, prepared figures and/or tables.

Jian Gao and Siqi Wang performed the experiments, contributed reagents/materials/analysis tools.

Pei-Chun Liao conceived and designed the experiments, analyzed the data, prepared figures and/or tables, authored or reviewed drafts of the paper, approved the final draft.

The following information was supplied relating to field study approvals (i.e., approving body and any reference numbers):

Field experiments were approved by the Honghuaerji Nature Reserve (permit number 200/66150221).

The following information was supplied regarding data availability:

The rarefied OTU table was generated using Qiime (Caporaso et al., 2010). All of the R code, rarefied OTU table and input files are available at Mendeley:

Huang, Bing-Hong (2018), “Afforestation effects on Mongolian Pine soil microbiomes”, Mendeley Data, v1, http://dx.doi.org/10.17632/gjskh8wswz.1.

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
