# Peer review of "The effects of afforestation on soil bacterial communities in temperate grassland are modulated by soil chemical properties"

_PeerJ, doi:10.7717/peerj.6147_

## Round 0.1 · original submission · Major Revisions

After my own reading of the manuscript and with two expert review assessments it appears you have identified an interesting system and provide potentially novel results. However, the manuscript has many deficiencies. In the methods it is not explained how soils were collected and the method explains that only soil below the root zone was used for analyses. However in other parts of the manuscript it seems that samples included the soil column. A detailed description of sample collection needs to be included in the revised submission.

Another issue is with the definition of the replication unit and if site is the unit of replication I am afraid there will not be enough degrees of freedom to draw any conclusions. The study maybe pseudo-replicated but it is hard to tell from the description of methods. Please clarify.

The description of the microbial data analyses in methods and beyond is confusing and not at all transparent. It is hard to assess its adequacy with the information given. Please be thorough with this aspect of bioinformatics.

Finally, the manuscript is on the long side. Parts of the intro and discussion could be shortened considerably. I suggest to eliminate all especulation (including some in the methods) which should help with streamlining the manuscript.

Please note that one of the reviewers made all the comments ion the pdf. When sending the rebuttal these comments must be included in your letter. Hope you find these comments useful.

·

Basic reporting

The following comments pertain to the above mentioned manuscript, in which Wu et al. examine the effects of afforestation on belowground bacterial communities and soil physicochemical properties. Specifically, this manuscript is written within the context of forest conservation efforts in China which aim to reestablish forested areas in regions that historically were not forests, but grasslands. Wu et al. compare soils in a 27-33 year old artificial forested area to the surrounding native grassland soils, and explore the relationships between soil chemistry and bacterial community structure and estimated function.
The authors take a unique approach in their analysis, which sets it apart from other literature using 16S rRNA data to examine soil communities. From this perspective, I found this manuscript to be rather refreshing, and I think it has potential to be a very interesting paper. However, there is much that needs to be addressed or improved before publication. Specific comments may be found in the annotated PDF submitted along with this review. Broader comments may be found in the following sections, as laid out per the PeerJ editorial submission page.



Clear and unambiguous, professional English used throughout:

For the most part, the language is acceptable. However, there are sections / sentences / phrases scattered throughout the manuscript that are unclear or vague in their point or meaning (see annotated PDF). While I only marked a couple of them, this would need to be addressed much more thoroughly before publication.
I also recommend a thorough check for the use of the word “microbial”, “microbiome”, or similar. While appropriate in some contexts within this manuscript, there are many where use of this verbiage is not. These words refers both to bacterial (prokaryotic) and fungal (eukaryotic) communities in soils, and the scope of this paper does not extend beyond bacterial community analysis (16S rRNA), and therefore the authors need to be more careful when using this term.


Literature references, sufficient field background/context provided:

There are a few confusing sections in the Introduction which are highlighted in the annotated PDF, but otherwise the authors do a good job providing background material and context for the study. The literature referenced in this study seems appropriate.


Professional article structure, figs, tables. Raw data shared:

The structure of the manuscript is appropriate. The figure and table captions need more thorough descriptions.
One of my major complaints about reviewing this manuscript is the inadequacy of the shared raw data processing and statistical analysis files. While the authors do provide a spreadsheet which contains counts of sequences at the OTU and phylum level, and subsequent COG and KEGG estimated functional gene counts, as well as reference to where the raw sequence data was uploaded to NCBI GenBank, they fail to provide the Unix code or python and R scripts used for bioinformatics, sequence read filtering, or statistical analysis, making it very difficult to determine exactly what was done, and impossible to check their work or verify their results.


Self-contained with relevant results to hypotheses:

This manuscript does contain all data that is relevant to the goal of the paper. If anything, certain aspects of the analysis may be unnecessary, or not directly address the questions put forward by the hypotheses. Specific recommendations may be found in the annotated PDF. But briefly, I recommend that prior to publication, the authors consider removing certain portions of the analysis that may be convoluted or unnecessary, and explore alternative, more direct approaches to answer the questions they pose.

Experimental design

Original primary research within Aims and Scope of the journal:

If sufficiently improved, this manuscript would fit in well with the Aims and Scope of PeerJ.


Research question well defined, relevant & meaningful. It is stated how research fills an identified knowledge gap:

The main research question posed in this manuscript (i.e. What is the effect of afforestation on soil physicochemical properties and bacterial community structure and function?) are relevant and meaningful to the topic introduced in the Introduction. Efforts to “re-forest” regions of China may have unintended environmental consequences, or be unsuccessful, if not done properly. As such, this research highlights potential consequences of such actions, while simultaneously exploring above- belowground interactions, relationships, and responses which are an important area of current research.
The hypotheses would benefit from being more well defined and direct, by stating what results would support or refute the hypotheses. Currently they are:
(1) The soil prokaryotic microbiome is indirectly affected by afforestation mediated by soil physicochemical properties, and
(2) Soil microbial ecofunctions are more sensitive to changes in vegetation type than changes in soil microbial composition.
In the case of (1), it will be difficult to distinguish if changes seen in the bacterial community are the result of soil physicochemical properties, or instead direct influence of Mongolian pine root exudation (NOT mediated by changes in soil properties)?
For (2), I am confused as to how this can be tested with the data gathered considering that the microbial ecofuncional data obtained from PICRUSt is directly derived from the microbial composition data?
Overall, while these hypotheses are relevant to the research question, they are vague as to what is actually being tested. It is unclear whether the main focus of this manuscript is to identify differences between the two vegetation types (grassland vs. forest), or if it is to examine the relationships between the soil environment and the microbial community, and subsequent functional capacities.


Rigorous investigation performed to a high technical & ethical standard:

The overall design of this experiment (10 soils from grassland vs. 10 soils from “forest”) is adequate to answer the research question. Although there are a few details that require further description (see below), from what I can tell from what is reported, the authors used proper laboratory techniques.
I would like to reiterate that the usage of PICRUSt is merely an estimation of functional gene abundance, which has not been fully vetted for use in soil environments where there are a great many unknown (unsequenced) and unculturable bacteria. I recommend that authors acknowledge this somewhere in the manuscript, and use the word “estimated” when referring to ecofunctional data. Also, be clear that these are functional genetic potentials (or capacities) as opposed to actual measurements or activity. See line 332 for example.


Methods described with sufficient detail & information to replicate:

There are many details about the collection (amount of soil collected, equipment used, sterilization techniques, etc.) which are not given in the Methods section (lines 137-144).
I also have some concerns about the depth of the soil sampling (50cm if I interpreted the methods correctly). Bacterial communities at 50cm soil depth will be very different from communities in shallower soils where much of the litter decomposition, gas exchange, and nutrient transformations occur. I fear that the soil samples which were collected do not fully represents the microbial communities in these plots. While I realize this cannot be corrected, I think it is important to acknowledge this in the methods and discussion sections, or reframe the paper to look only at bacterial communities within a certain soil layer/depth. It would also help to further characterize the soil profile by identifying the soil layers (O, A, E, B, horizons) and reporting which horizon occurs at 50cm.
The authors did a very good and thorough job of reporting DNA extraction and sequencing procedures.
While the actual descriptions of the statistical analyses are adequately described, the authors do not report how the analyses were computed (what software was used). If done in R, I would greatly appreciate seeing the code that was used. Also, there were a few inconsistencies in which tests were used (line 196 vs. figure 1), and some confusion as to what analyses data transformations were used on (methods state that all abundance counts were transformed to relative abundance, but Fig. 4 shows count abundance data).

Validity of the findings

Impact and novelty not assessed. Negative/inconclusive results accepted. Meaningful replication encouraged where rationale & benefit to literature is clearly stated:

No comment.


Data is robust, statistically sound, & controlled:

Without more descriptive specifics on how the data was handled and analyzed, it is difficult to judge how sound the analyses are. However, the data that is reported seems reasonable.


Conclusion are well stated, linked to original research question & limited to supporting results:

The conclusions, while a bit speculative, seem sound. They are linked to the original research question, but fail to identify key specifics, such as what the potential functional differences are between grassland plots and afforested plots. The analysis reveals that there is a difference in overall genetic function potential, but doesn’t specify what functional genes are driving this difference. As such, the authors can not speculate on how afforested regions which were once grasslands might respond functionally to this “disturbance”.


Speculation is welcome, but should be identified as such:

The language used does a proper job of identifying speculation.

Additional comments

First, thank you for your submission! I greatly appreciate multiple aspects of this manuscript, including not only the context within which it is set, but the unique approach you take with the analysis. I recognize the context of afforestation efforts in China to be of great environmental concern, and research such as this is important in understanding the impacts of artificially manipulated natural areas, even if the original intent is based in conservation efforts. Also, your statistical analysis of this data is impressive. I appreciate your effort to use a variety of statistical methods to untangle the complex interactions inherent in belowground ecosystems.
There are number of things that can be done to improve this manuscript. Much of detailed comments can be found in the annotated PDF which I will submit with this review, but the following are the most important:
1) Improve details in Methods. More detailed site information as it pertains to soil characteristics and natural history of the region. More detailed soil characterization and collection description (what horizon was the soil collected from at 50cm). Provide additional information / raw data on how the analyses were done, including any bioinformatics pipelines, Unix code, R code, QIIME scripts, etc. The statistical analysis should be able to be repeatable from the raw data files submitted.
2) Focus hypotheses. Make sure they are testable and that the data you have gathered can test them.
3) Focus analysis. Once you have testable hypotheses, make sure that the statistical tests you use are directly related to testing them. You use many different analyses, some of which I think are great and useful and informative (cluster analysis, DAPC, dbRDA, CLAM), but others seem to confuse (multiple SLR and MLR models, Rand Abundance Dominance, etc.). Consider using other methods as well (e.g. PERMANOVA, Mantel tests, differential abundance analysis (DESeq2)). Be careful not to exclude important players that are most likely biologically important (i.e. most abundant phyla such as Acidobacteria, Verrucomicrobia, or Proteobacteria, and soil chemical factors such as pH).
4) Focus writing. There are many instances where the text is confusing, vague, or contradictory. Be sure what you are trying to say is clear and has a purpose.

·

Basic reporting

1.Clear and unambiguous, professional English used throughout.
The presented manuscript has been written in correct English, is clear to understand, unambiguous, and technically correct.
2.Literature references, sufficient field background/context provided.
The manuscript includes a very nice introduction, prior literature and was mention new literature regarding the research topic.
3.Professional article structure, figs, tables. Raw data shared.
The manuscript structure is good. Tables and Figures have a good quality to publish in PeerJ journal. However, according to my comment (.pdf file attached), I would appreciate if the authors could include RDA plots to show the community structure. All appropriate raw data has been made available in accordance with PeerJ "Data Sharing policy".
4.Self-contained with relevant results to hypotheses.
The manuscript also seems to be ‘self-contained,’ and appropriate to ‘unit of publication’, and all results were relevant to the initial hypothesis.

Experimental design

1.Original primary research within Aims and Scope of the journal.
The manuscript is an original primary research and seems to be within Aims and Scope of the journal.
2.Research question well defined, relevant & meaningful. It is stated how research fills an identified knowledge gap.
The research question was defined and relevant.
They hypothesized that the soil microbiome assessed by 16S analysis is indirectly affected by afforestation mediated by soil chemical properties and soil microbial ecofunctions are more sensitive to changes in vegetation type than changes in soil microbial composition.
In a conclusion, they proved the initial hypothesis, and the change in vegetation type was linked, at least in part, to ecofunctional changes in the soil microbiome, despite an indirect impact on microbial composition. However, the authors should do some modifications in Conclusion to improve quality (please see .pdf file attached).
3.Rigorous investigation performed to a high technical & ethical standard.
The investigation has been conducted rigorously and to a high technical standard. The research has been conducted in conformity with the prevailing ethical standards in the field.
4.Methods described with sufficient detail & information to replicate.
In general, the M&M section is good. However, the authors should do some modifications to improve quality (please see .pdf file attached).

Validity of the findings

1.Impact and novelty not assessed. Negative/inconclusive results accepted. Meaningful replication encouraged where rationale & benefit to literature is clearly stated.
The manuscript has answered important questions about the effects of afforestation practice on soil microbiome. This type of practice can alter soil fertility, health, and to know the microbial response is very important to environmental sustainability and new research in future. The data is robust, statistically sound, & controlled.
2.Conclusion are well stated, linked to original research question & limited to supporting results.
The conclusions have been appropriately stated, connected to the original question investigated, and supported by the results. Please, see additional comments on .pdf file attached.

Additional comments

Dear authors, I have done specific comments in your manuscript submitted to PeerJ journal. You can find all comments assessing the .pdf file attached. I would be happy if you can make changes to improve your finds. I think after major revisions this manuscript has big potential to be published in PeerJ journal.

---

## Round 0.2 · Minor Revisions

Thank you for your revised manuscript. You have done a generally good job at addressing at the multiple concerns by reviewers but did not address those from the handling editor directly. I find the manuscript close to acceptance except that the English needs major improvements. Grammar and sentence construction do not conform to normal accepted English practices for scientific papers. Use of proper English is responsibility of the authors, but please take special attention to your revised added sections as they appear to be the most problematic. Thanks.

·

Basic reporting

None comments.

Experimental design

None comments.

Validity of the findings

None comments.

Additional comments

Dear authors.

Thank you for following carefully all initially requested suggestions. Thus, as per my first criterious review, my final decision is to accept this paper to publication in PeerJ journal.

With best regards.

---

## Round 0.3 · accepted · Accept

Thanks you for the quick turnaround and the corrections of the English. There are a couple of minor issues that need attention:

- for soil samples please indicate the surface area in cm2 or m2. Also indicate if soils were sampled from a pit or from a soil corer.

- "ecofunction" is not a word. Please use "ecological function", "ecological role" or just function/functional

Thanks

Miquel

#